# Current Status of Research on Wildland Fire Impacts on Soil Environment and Soil Organisms and Hotspots Visualization Analysis

Zhichao Cheng [1,2,†], Song Wu [3,†], Dan Wei [1,†], Hong Pan [1], Xiaoyu Fu [1], Xinming Lu [1] and Libin Yang [1,2,*]

1    Key Laboratory of Biodiversity, Institute of Natural Resources and Ecology, Heilongjiang Academy of Sciences, Harbin 150040, China; chengzc928@163.com (Z.C.); weidan_0929@163.com (D.W.); panhong500@163.com (H.P.); 18646583130@163.com (X.F.); luxinming0210@163.com (X.L.)
2    Heilongjiang Huzhong National Nature Reserve, Huzhong 165038, China
3    Science and Technology Innovation Center, Institute of Scientific and Technical Information of Heilongjiang Province, Harbin 150028, China; wusong0927@126.com
*    Correspondence: 13664600518@139.com
†    These authors contributed equally to this work.

**Abstract:** Ecosystems are frequently disturbed by fires that have an important impact on the soil environment and the composition of soil organisms. In order to provide a baseline for the current research and identify trends on the effects of wildland fire on soil environment and biological changes, the available literature was identified from the Web of Science database, covering the period from 1998/1998/1999 (the year of the earliest publication in this field) to 2023. A bibliometric analysis was performed and the data were visually displayed for the number of publications, countries, authors, research institutions, and keywords representing research hotspots. Specifically, the effects of wildland fire on the soil environment, on soil microorganisms and on soil fauna were analyzed. The results show that the annual number of publications describing effects of wildland fire on the soil environment and on soil microorganisms are increasing over time, while those describing effects on soil fauna are fewer and their number remains constant. The largest number of papers originate from the United States, with the United States Department of Agriculture as the research institution with the largest output. The three authors with the largest number of publications are Stefan H. Doerr, Manuel Esteban Lucas-Borja and Jan Jacob Keizer. The research hotspots, as identified by keywords, are highly concentrated on wildfire, fire, organic matter, and biodiversity, amongst others. This study comprehensively analyzes the current situation of the research on the effects of wildland fire on changes in the soil environment and organisms, and provides reference for relevant scientific researchers in this trend and future research hotspots.

**Keywords:** wildland fire; soil environment; soil organisms; bibliometrics

## 1. Introduction

Fire is a major disturbance factor for ecosystems and represents one of the main driving factors determining the structure and function of ecosystems [1]. Under the influence of global warming, wildland fires have increased in frequency, range, intensity, and severity, making their impact more complex. The impact of wildland fire on the soil environment depends on the local soil type and its moisture content, the intensity, frequency, and duration of the fire, and the type and quantity of combustible materials [2]. In general, low severity fires may only burn vegetation and some organic matter at the surface, while organic carbon in deeper soils may be relatively unaffected. Moderate severity fires may partially disrupt the soil structure, leading to decomposition and loss of organic carbon, whereas high severity fires may lead to more extensive soil burning, depleting the soil of large amounts of organic matter and reducing soil organic carbon content [3]. In addition,

fires may result in the loss of soil nutrients, such as nitrogen and phosphorus, affecting soil fertility and nutrient quality [4].

When fires occur, they not only have an impact on above-ground vegetation and the structure and function of the ecosystem, but also directly or indirectly affect the biology in the underground soil [5]. The high temperatures generated during a fire can be directly lethal to soil organisms [6] or, as an indirect effect, the loss of soil organic matter and nutrients results in the death of ecosystem members or their inability to survive [7,8]. The short-term impact of wildland fire on soil quality is usually negative, as it affects soil microorganisms that decompose organic matter and are responsible for nutrient cycling and nitrogen fixation, thereby regulating the soil carbon-to-nitrogen ratio [9]. Soil fauna can promote the recovery of an ecosystem after a fire by consumption of plant and animal remains, accelerating the turnover of microorganisms, and stimulating nutrient transformation [10].

Bibliometric analysis is a useful tool to analyze the research status in a certain field and identify current research hotspots and trends [11–13]. Through this method, the literature data of a target field can be quantitatively analyzed to obtain an overview of the current research situation [14,15]. There is a large body of literature that describes the impact of wildland fire on the soil from various parts of the world. However, at present a bibliometric analysis on the impact of wildland fire on soil has not been performed. This knowledge gap was filled here: the gradual increase in knowledge was assessed for the interaction between wildland fire and the ecological environment. We used bibliometrics to analyze the research dynamics and explored the research hotspots and development trends in soil environment and soil organisms after fires, in order to provide research directions and data for future research. After collecting publications describing the soil environment and soil organisms after wildland fire, bibliometric analysis identified the hotspots and dynamics of the current research on the impact of fire on the soil environment and organisms by analyzing the high-frequency publication volume, countries, institutions, authors, and keywords in the field. This provides references for scholars to conduct future research in this area.

## 2. Data Sources and Methods

VOSviewer is a very useful tool for mapping scientific knowledge and is widely used to visualize the available knowledge in a given domain. We used the Web of Sciences TM core set data (source: the Web of Sciences database) to conduct an advanced search of foreign literature using the following search terms: (forest p/0 fire or fogo p/0 selvagem or wildfire$ or bush p/0 fire or prairie p/0 fire or prescrib* p/0 fire or plan* p/0 fire or fire p/0 treat* or fire p/0 disturb* or fire p/0 regime) and (soil p/2 condit* or soil environ* or soil chemic* or soil physic* or soil p/2 propert*), (forest p/0 fire or fogo p/0 selvagem or wildfire$ or bush p/0 fire or prairie p/0 fire or prescrib* p/0 fire or plan* p/0 fire or fire p/0 treat* or fire p/0 disturb* or fire p/0 regime) and (soil microbe* or soil bacteria* or soil fung*), (forest p/0 fire or fogo p/0 selvagem or wildfire$ or bush p/0 fire or prairie p/0 fire or prescrib* p/0 fire or plan* p/0 fire or fire p/0 treat* or fire p/0 disturb* or fire p/0 regime) and (soil animal* or soil fauna*), with the literature type "Topic". The data were saved as plain text records, and their contents were selected as full records and cited references. Excel was used to create relevant tables and VOSviewer 1.6.20 was used to visualize the hotspots of countries, institutions and keywords [16–19].

## 3. Results

We identified publications on the impact of wildland fire covering the period from 1998/1998/1999 (the year of the first publication in the field) to 2023. A total of 2388 literature records were collected based on our searches. These were analyzed for three subsets, with literature describing the effects of fires on the soil environment ($n$ = 1770), publications related to the effects on soil microorganisms ($n$ = 479) and literature on the effects of wildland fire on soil fauna ($n$ = 139). Their separate analyses are presented here for bibliometric characteristics (annual publication volume, country, institution, author) and

for keyword analysis. Each section starts with a brief state-of-the-art summary, based on the body of literature.

### 3.1. Current Status of Research on the Impact of Wildland Fire on the Soil Environment

#### 3.1.1. State of the Art

It has been shown that low and moderate intensities of fire will not directly affect the soil structure whereas, under severe fire generating high temperatures, organic matter and roots will decompose, exposing the inorganic soil, which alters the soil structure and results in soil compaction and hardening, thereby reducing the air permeability and water retention of the soil [20]. Rainfall is the primary driving force of soil erosion [21]. Vegetation and litter function to intercept and absorb rainwater [22,23]. After a fire, the impact of rainwater on the surface and surface runoff will be reduced, thereby reducing soil erosion [24]. In addition, soil pH has a significant impact on plants and soil organisms. The ashes of plants and their dead organisms usually contain large amounts of $Mg^{2+}$, $K^+$, and $Ca^{2+}$, etc., which can increase the pH of the soil in the fire area [25,26]. Hamman et al. [27] found that, as the intensity of the fire increased, the soil pH also significantly increased; however, under severe fire intensity, the soil pH was significantly lower than that of an unburned control site, due to the consumption of soil nutrients [28]. Wildland fire can affect the bio-geochemical cycling of elements such as C, N, and P in the soil [29]. After a fire, the loss of surface soil and surface litter layer, caused by surface runoff, results in different degrees of loss, from slight distillation (volatilization of minor components) to carbonization or complete oxidation, resulting in a reduction in the content of soil carbon and nitrogen [30,31]. In addition, during the fire, C and N bound in complex organic matter are converted to inorganic compounds (such as $CO_2$ and $NO_x$), and these chemically react with the soil, thereby affecting its acidity and fertility [32,33]. After the fire, P in the ground cover and other combustible materials volatilizes and leaches in the form of fine ash particles, but the impact of wildland fire on P is not as significant as on N [25,34].

#### 3.1.2. Annual Publication Volume, Countries, Institutions, and Authors

The annual volume of publications over time identifies any changes in the number of documents in a certain research field [35,36]. The 1770 publications describing the effects of fires on the soil environment were binned for year of publication. As can be seen from Figure 1, the annual number of publications on this subject increased over time. The plot suggests that research on the impact of wildland fire on the soil environment has further potential to grow.

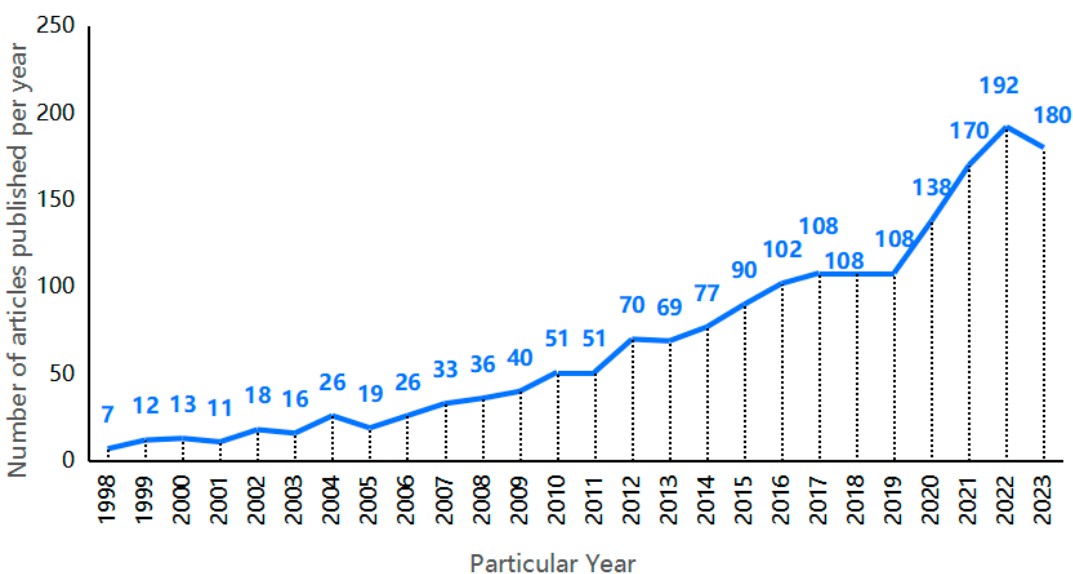

**Figure 1.** Number of annual publications describing effects of wildland fire on the soil environment.

The number of publications from a country is taken as a measure to represent the degree of activity and contribution of that country in the research field [37,38]. Of the 1770 documents retrieved by the searches, 672 were from USA, followed by Spain (333), Italy (101), and Australia (41). The data were analyzed for national co-occurrence and summarized in a graph (Figure 2), which illustrates that the USA, Spain, England and Australia were most closely associated.

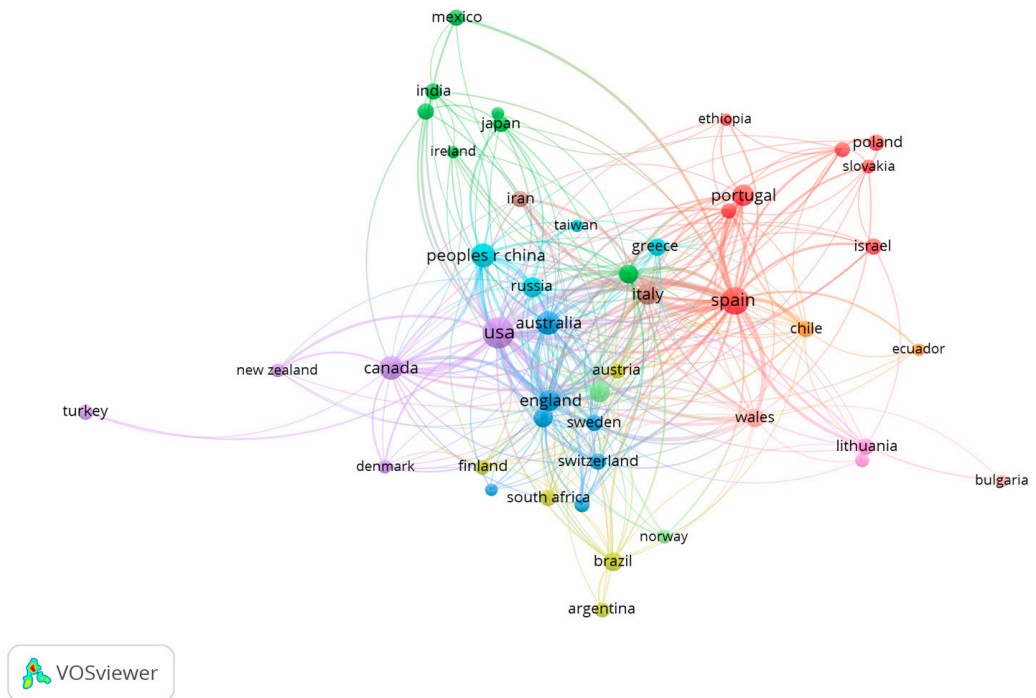

**Figure 2.** National co-occurrence mapping of 1770 publications describing effects of wildland fire impacts on the soil environment.

The top three research institutions responsible for the highest numbers of publications were identified as the United States Department of Agriculture (157), United States Forest Service (116) and United States Department of Interior (101). Table 1 summarizes the 10 top-scoring organizations that together accounted for 46.5% of the total number of publications. The research institutions located in the United States are more productive than those in other countries in the field of soil environmental effects in the aftermath of wildland fire.

**Table 1.** Research institutions producing the largest number of publications on the impact of wildland fire on the soil environment.

| Affiliation | Record Count | Affiliation | Record Count |
|---|---|---|---|
| United States Department of Agriculture | 157 | Consejo Superior de Investigaciones Cientificas | 87 |
| United States Forest Service | 116 | Chinese Academy of Sciences | 49 |
| United States Department of Interior | 101 | University of Idaho | 46 |
| United States Geological Survey | 96 | Russian Academy of Sciences | 42 |
| University of California | 88 | Swansea University | 41 |

Co-occurrence analysis of the authors of published literature provides a good indication of the degree of importance of the core authors in a given research field, whereby different colors represent different groups of authors [39,40]. A graph representing these co-occurrence statistics for publications describing the effects of wildland fire on soil environment is shown in Figure 3. The top five authors in terms of number of publications were, in order, Stefan H. Doerr (Swansea Univ, Sch Environm and Soc, Dept Geog. Wales. 35 articles), Manuel Esteban Lucas-Borja (Univ Castilla La Mancha, Dept Agroforestry Technol

and Sci and Genet, Sch Agr and Forestry Engn. Albacete. 35 articles), Jan Jacob Keizer (Univ Aveiro, Dept Environm. Aveiro. 30 articles), Paulo Pereira (Univ Barcelona, Dept Phys Geog and Reg Geog Anal, GRAM Mediterranean Environm Res Grp. Catalunya.; Mykolas Romeris Univ, Dept Environm Policy and Management. Vilnius.; Vilnius Gedminas Tech Univ, Environm Protect Dept. Vilnius. 22 articles) and Jorge Mataix-Solera (Univ Miguel Hernandez, Dept Agrochem and Environm, Environm Soil Sci Grp. Alicante. 21 articles) The distribution of authors is characterized by a small concentration and a large overall dispersion (Figure 3).

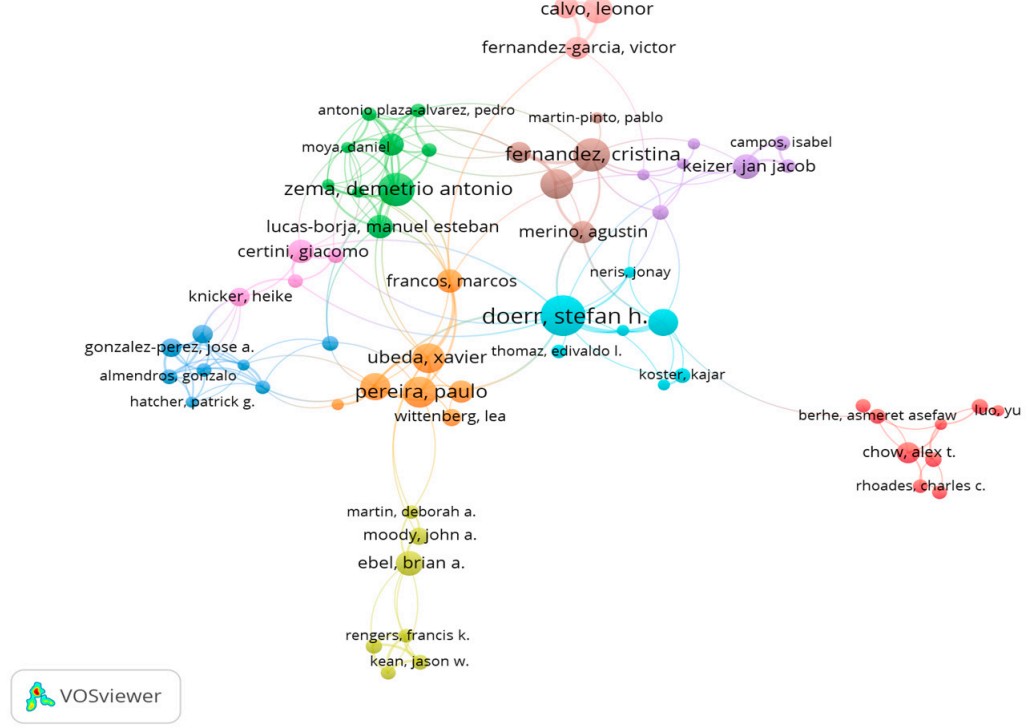

**Figure 3.** Author co-occurrence mapping of publications describing effects of wildland fire effects on the soil environment.

### 3.1.3. Keyword Co-Occurrence and Clustering

The keywords of the 1770 samples of literature data were collected, which resulted in a total of 8237 keywords. These were statistically analyzed, and their relatively frequency of occurrence was determined. Those keywords that appeared at least five times were considered high-frequency keywords, which applied to 764 keywords. The frequency ranking revealed that 10 keywords appeared 159 times or more, indicating that these keywords have received most attention in research on the impact of wildland fire on the soil environment (Table 2).

**Table 2.** Top 10 keywords with the highest occurrence frequency in the literature on the impact of wildland fire on soil environment.

| No. | Keyword | Occurrence | No. | Keyword | Occurrence |
|---|---|---|---|---|---|
| 1 | Wildfire | 695 | 6 | Vegetation | 184 |
| 2 | Fire | 502 | 7 | Climate Change | 173 |
| 3 | Organic matter | 248 | 8 | Carbon | 172 |
| 4 | Soil | 214 | 9 | Erosion | 162 |
| 5 | Nitrogen | 185 | 10 | Forest | 159 |

The 764 identified high-frequency keywords were analyzed for co-occurrence (Figure 4). The larger the circles in the figure, the more frequent that keyword appeared. Such keyword co-occurrence and cluster analysis can reveal hot topics and trends in a given research field, and evaluates the quality and impact of research results, whereby different colors represent different clusters [41,42]. From Figure 4, it can be seen that the keywords separate into six clusters (indicated by colors), indicating that, from 1998 to 2023, there were six important research topics in the field of the impact of wildfires on soil environment: "Wildfire", "Organic matter", "Vegetation", "Black carbon", "Climate Change" and "Burn severity". The results indicate that the current focus is mainly on four aspects: (1) effects of fire on nutrient content, transformation and cycling in soils, and how soil fertility can be improved through fire management [43,44]; (2) the process of restoring ecosystems after fires, including revegetation, soil remediation and restoration of ecological functions [45]; (3) studies towards the effects of climate change on the frequency and intensity of fires and how fires in turn affect the climate system [46]; and (4) analyses of the impact of fire on soil erosion and water erosion and what measures can be taken to mitigate these problems [47].

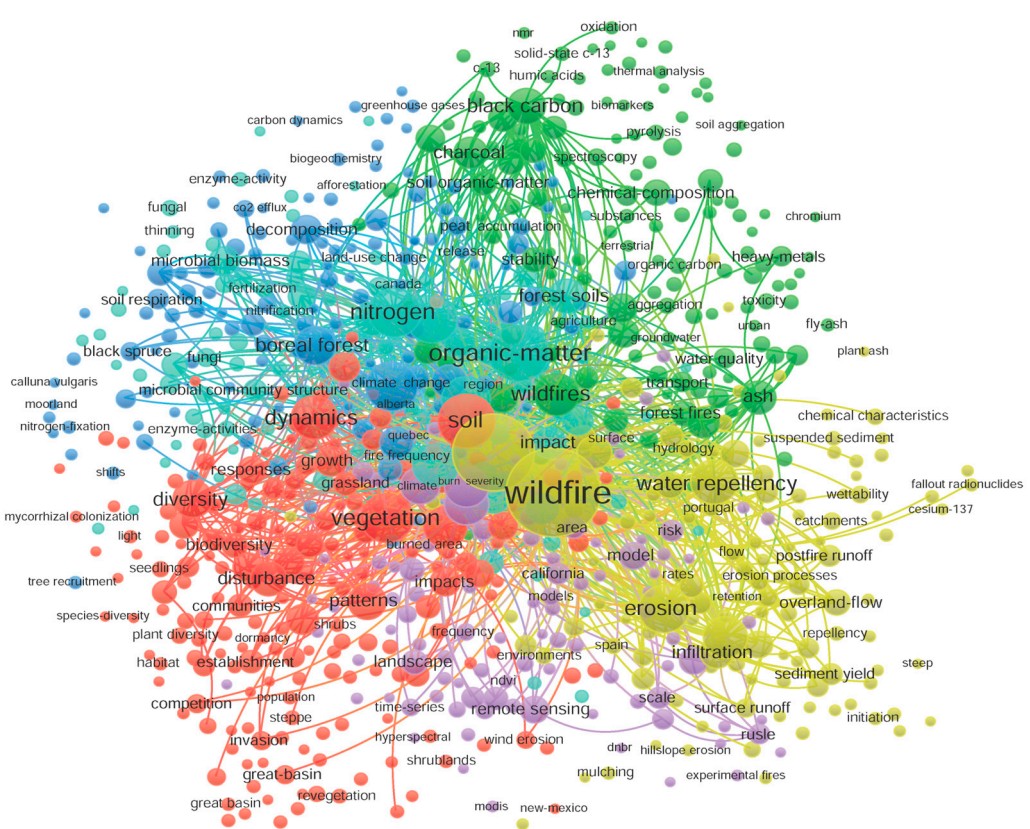

**Figure 4.** Co-occurrence analysis of high frequency keywords in publications describing effects of the impact of wildland fire on soil environment.

### 3.2. Current Research Status of the Impact of Wildland Fire on Soil Organisms

#### 3.2.1. State of the Art

The impact of fire on soil microorganisms has two main aspects: first, the high temperature generated by wildfires directly kills soil microorganisms and, second, wildfires change their environment, indirectly affecting their growth and abundance [6,7]. Fire intensity, soil moisture content, soil pH, and available nutrients can all affect the composition and quantity of soil microorganisms [48]. High intensity fire can kill microorganisms by rupturing cell membranes, denaturing proteins and destroying nucleic acids [49]. Wu et al. found that a decrease in soil microbial biomass correlated with increasing fire intensity ($p < 0.05$) [50], but contrasting results have also been reported; for example, Fultz et al. described that wildfires significantly increased the soil microbial biomass ($p < 0.05$) [51].

This may be because fire can accelerate the process of decomposition of dead biomatter, thereby increasing the availability of effective nutrients in the soil and promoting microbial growth and metabolism [52]. Bacteria and fungi are the two main microbial taxa present in soil ecosystems, but their viability strategies and ecological functions differ, resulting in differences in their ability to resist high temperatures [53,54]. In general, fungi are more sensitive to high temperatures than bacteria; in addition, wildfires increases soil pH and the abundance of light areas increases, while aeration decreases. In combination, this provides more suitable conditions for bacterial growth than for fungal growth. The faster growth rate of bacteria also allows for a faster recovery, which may explain why autotrophic, anaerobic, nitrifying, and nitrite bacteria are usually present in higher numbers in the soil after a fire than in control soil [55,56]. On the contrary, fungi are very sensitive to fire, and their diversity usually decreases after fire [57]. However, Cheng et al. found that the relative abundance of arbuscular mycorrhizal fungi recovered to pre-fire levels following high intensity fires disturbance ($p > 0.05$) [58].

### 3.2.2. Annual Publication Volume, Countries, Institutions, and Authors

A total of 479 publications from 1998 to 2023 could be identified that dealt with the impact of wildland fire on soil microorganisms, with an upward trend for number of publications per year, in particular since 2018 (Figure 5). Although the research on effects on soil microorganisms caused by wildland fire started relatively late and is currently still in an early stage of maturity, global attention is increasing for wildland fire ecology and the restoration of post-fire ecosystems. It can be expected that the number of annual publications will continue to increase in the future.

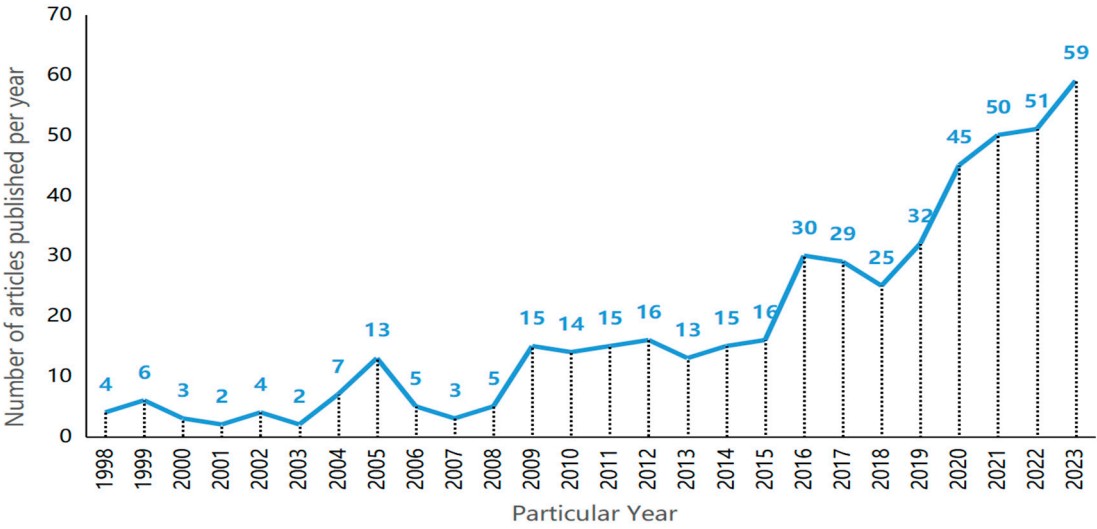

**Figure 5.** Number of annual publications describing effects of wildland fire effects on soil microorganisms.

Figure 6 summarizes the national co-occurrence mapping of the relevant publications, with the USA and Spain as central nodes, indicating that these countries have strongly contributed to this field of research. This may also be because these highly developed countries enable relatively extensive scientific exchanges and collaborations.

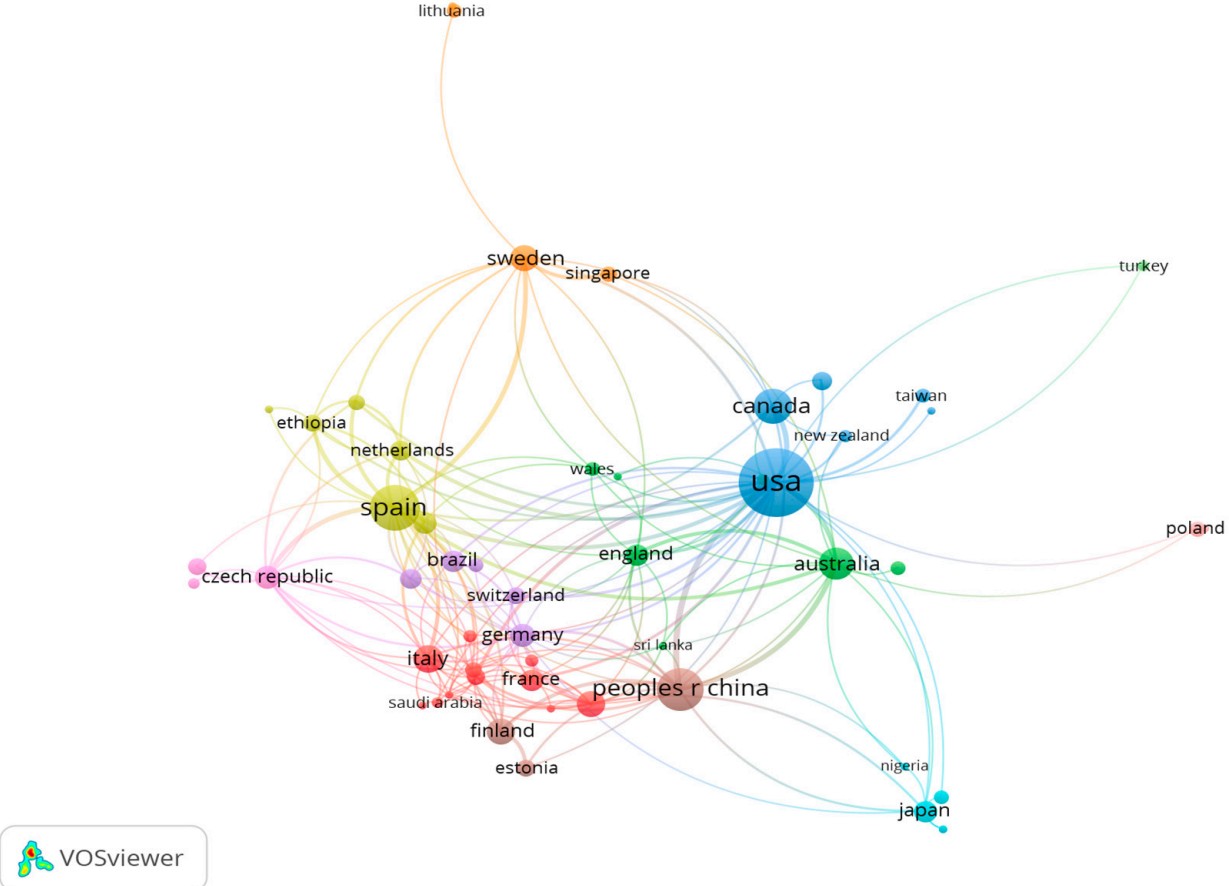

**Figure 6.** National co-occurrence mapping of 479 publications describing wildland fire impacts on soil microorganisms.

The top three institutions in terms of publication volume in this field are the University of California (45 articles), the United states Department of Agriculture (40 articles), and the United States Forest Service (34 articles) (Table 3). The 10 most productive institutes are responsible for 51.36% of the total number of identified publications, indicating their high level of activity in this field and their playing of an important role in describing the response mechanism of wildland fire to soil microorganisms.

**Table 3.** Research institutions with the highest output describing the impact of wildland fire on te soil microorganisms.

| Affiliation | Record Count | Affiliation | Record Count |
|---|---|---|---|
| University of California | 45 | Oregon State University | 18 |
| United States Department of Agriculture | 40 | United States Department of Energy | 18 |
| United States Forest Service | 34 | Chinese academy of Sciences | 16 |
| Consejo Superior de Investigaciones Cientificas | 27 | Swedish University of Agricultural Sciences | 15 |
| Universidad de Valladolid | 19 | Czech Academy of Sciences | 14 |

The five authors with the highest number of publications were Pablo Martin-Pinto (Univ Valladolid Palencia, Fire and Appl Mycol Lab, Dept Ciencias Agroforestales Prod Vegetal and Recur, Sustainable Forest Management Res Inst. Palencia. 18 articles), Juan Andres Oria-de-Rueda (Univ Valladolid Palencia, Fire and Appl Mycol Lab, Dept Ciencias Agroforestales Prod Vegetal and Recur, Sustainable Forest Management Res Inst. Palencia. 16 articles), Kajar Köster (Univ Helsinki, Dept Forest Sci; Estonian Univ Life Sci, Inst

Forestry and Rural Engn. Helsinki. 11 articles), and Frank Berninger and Jukka Pumpanen (both Univ Helsinki, Dept Forest Sci. Helsinki. 10 articles). The co-occurrence graph of authors (Figure 7) illustrates how authors cooperated in different clusters in their respective research areas, but the density of connections between each cluster is relatively small, and the cooperation between different teams is relatively loose.

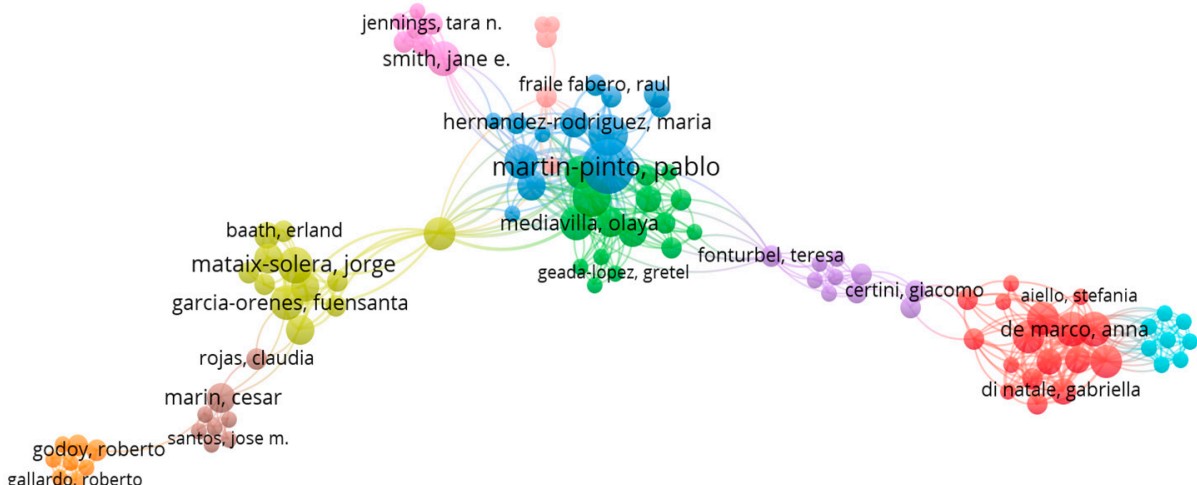

**Figure 7.** Author co-occurrence for publications dealing with wildland fire effects on soil microorganisms.

3.2.3. Keyword Co-Occurrence and Clustering for Wildland Fire Effects on Soil Microorganisms

From the 479 selected publications, a total of 2564 keywords were extracted, of which 202 were used at least five times. The top 10 keywords appeared at a frequency of at least 60 (Table 4).

**Table 4.** Top 10 keywords with highest occurrence frequencies for impact of wildland fire on soil microorganisms.

| No. | Keyword | Occurrence | No. | Keyword | Occurrence |
|-----|---------|-----------|-----|---------|-----------|
| 1 | Wildfire | 214 | 6 | Soil | 69 |
| 2 | Fire | 182 | 7 | Nitrogen | 69 |
| 3 | Diversity | 130 | 8 | Dynamics | 65 |
| 4 | Organic matter | 82 | 9 | Forest | 62 |
| 5 | Carbon | 81 | 10 | Biomass | 60 |

From Figure 8, it can be seen that the keywords co-occur in six clusters around the research topics "Wildfire", "Fire", "Boreal forest", "Organic-matter", "Temperature", and "Fungal communities". The focus of this literature was mainly on changes in the composition of soil microbial communities after fire, and their impact on soil ecosystem functioning, indicating in what processes soil microorganisms play a key role during the recovery of an ecosystem after fire [59,60].

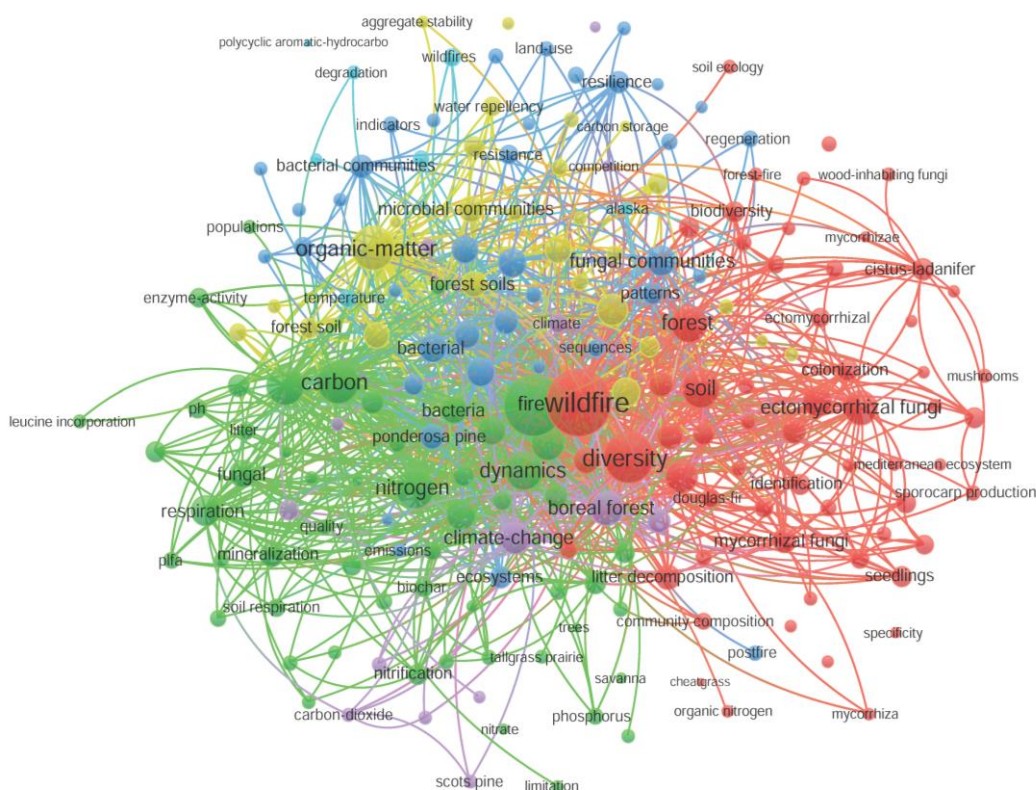

**Figure 8.** High frequency keywords co-occurrence in publications on the impact of wildland fire on soil microorganisms.

### 3.3. Current Research Status of the Impact of Wildland Fire on Soil Fauna

#### 3.3.1. State of the Art

The last subject covered in this bibliometric analysis was the impact of wildland fire on soil fauna. Fires are typically lethal for animals living in the top layer of the dead branches and leaves of the soil, but animals living in the semi-decomposed layer and in the humus layer may survive [61]. Fires typically result in an irregular distribution pattern for most of the surviving soil fauna, forming an island-like spatial distribution pattern with a small number of areas where the soil fauna survives. Their occurrence may depend on the extent of plant and tree cover, a variation that leads to spatial heterogeneity [62,63]. The soil fauna is dominated by nematodes, accounting for over 89% of all animals [61], and a fire can cause severe changes in the abundance of the main nematode groups, although one study reported no significant differences in the number of genera detected in burned and pristine soil samples [64,65]. A study describing that environmental conditions, which obviously change after a fire, have an impact on the activities and community characteristics of soil animals. Soil temperature plays a dominant role in the population density of nematodes, while their response to soil moisture levels is insignificant [66,67]. Although fire disturbance can alter the composition and spatial distribution of soil animal groups, such disturbances typically have a short duration with a fast recovery rate [61].

#### 3.3.2. Annual Publication Volume, Countries, Institutions, and Authors

Fewer publications dealt with the effect of wildland fire on soil fauna compared to the effects on soil environment or microorganisms: we were able to identify only 139 publications, and the yearly output is relatively constant (Figure 9).

This field is not really dominated by contributions from a particular country, as Figure 10 illustrates, although publications from USA and Australia are slightly overrepresented.

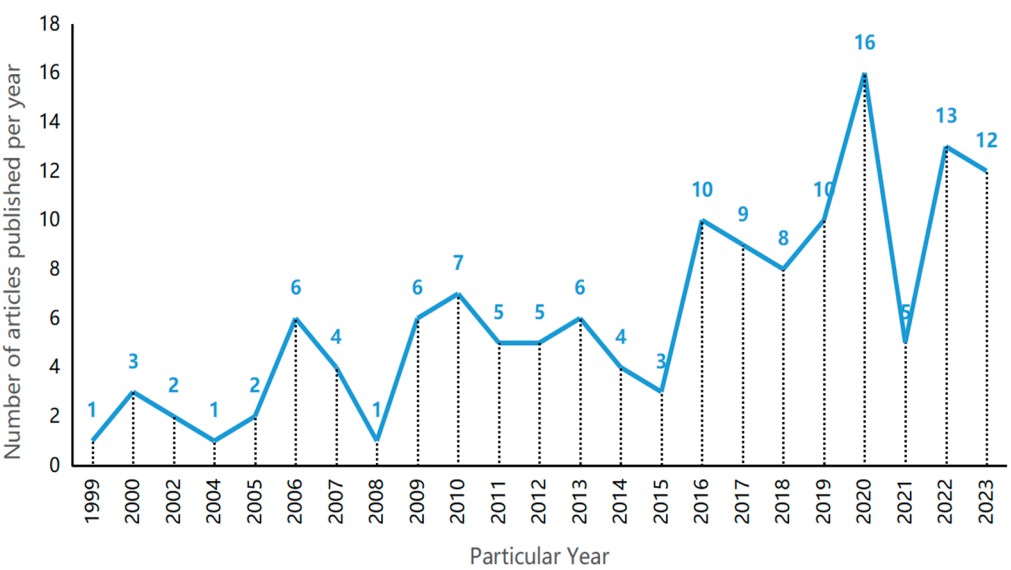

**Figure 9.** Number of annual publications describing effects of wildland fire effects on the soil fauna.

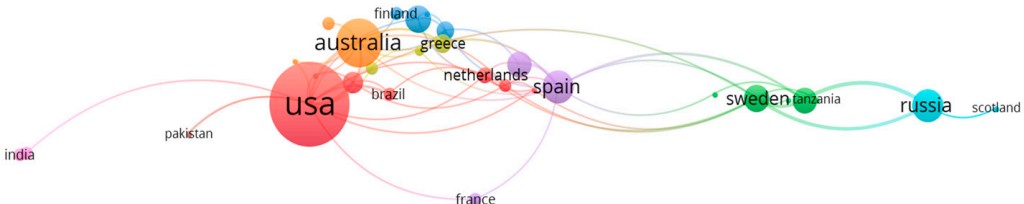

**Figure 10.** National co-occurrence mapping of 139 publications on impact of wildland fire on soil fauna.

The two institutions responsible for the most publications were the United States Department of Agriculture and the Russian Academy of Sciences (both 12 articles), as shown in Table 5. The research results indicate that, compared to these countries, the Asian region clearly pays less attention to investigation of the impact of wildland fire on soil animals.

**Table 5.** Research institutions with the highest output on the impact of wildland fire on fauna.

| Affiliation | Record Count | Affiliation | Record Count |
|---|---|---|---|
| United States Department of Agriculture | 12 | Swedish University of Agricultural Sciences | 7 |
| Russian Academy of Sciences | 12 | United States Department of Interior | 6 |
| Severtsov Institute of Ecology and Evolution | 8 | United States Geological Survey | 6 |
| Saratov Scientific Center of the Russian Academy of Sciences | 8 | Australian National University | 6 |
| United States Forest Service | 8 | Justus Liebig University Giessen | 5 |

The most productive authors in this subject were Konstantin B. Gongalsky (Russian Acad Sci, AN Severtsov Inst Ecol and Evolut. Moscow. 8 articles), Andrey S. Zaitsev (Russian Acad Sci, AN Severtsov Inst Ecol and Evolut. Moscow.; Justus Liebig Univ, Inst Anim Ecol. Giessen. 6 articles), and Jan Bengtsson and Tryggve Persson (both from Swedish Univ Agr Sci, Dept Ecol. Uppsala. 4 articles). Clearly, authors from Sweden are over-represented in this list. The co-occurrence graph of authors (Figure 11) identifies two clusters, among which Bengtsson has the most correlations with other teams.

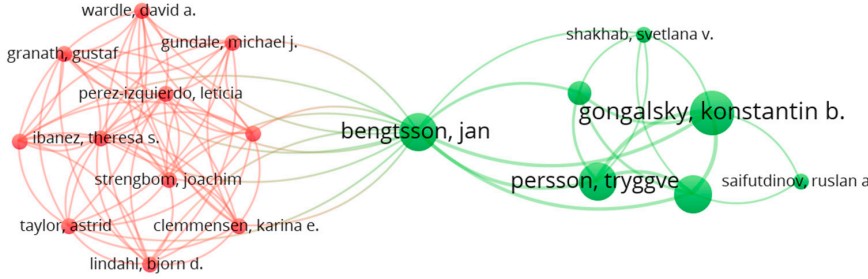

**Figure 11.** Author co-occurrence mapping for publications on wildland fire effects on soil fauna.

### 3.3.3. Keyword Co-Occurrence and Clustering for the Impact of Wildland Fire on Soil Fauna

Related to the smaller data set of publications, only 1163 keywords were identified. Among them, 46 appeared with a frequency of at least 5, and the top ten most frequently ranking keywords appeared at least 13 times (Table 6).

**Table 6.** Top 10 keywords with highest occurrence frequencies for impact of wildland fire on soil fauna.

| No. | Keyword | Occurrences | No. | Keyword | Occurrences |
|-----|---------|-------------|-----|---------|-------------|
| 1 | Wildfire | 51 | 6 | Disturbance | 16 |
| 2 | Fire | 47 | 7 | Diversity | 15 |
| 3 | Vegetation | 21 | 8 | Management | 15 |
| 4 | Biodiversity | 20 | 9 | Carbon | 13 |
| 5 | Response | 17 | 10 | Climate Change | 13 |

The co-occurrence analysis for keywords is shown in Figure 12, which contains four clusters, around "Wildfire", "Fire", "Management" and "Diversity". These summarize the current focus of the available literature on the characteristics of soil animal diversity, spatial and temporal distribution of soil fauna, and individual density clustering of soil animal communities after wildfires. This underlines that soil animal communities are an important component of ecosystems, where they are a crucial part of the underground debris food network and, as such, soil animals are essential for the restoration and reconstruction of burned areas [68].

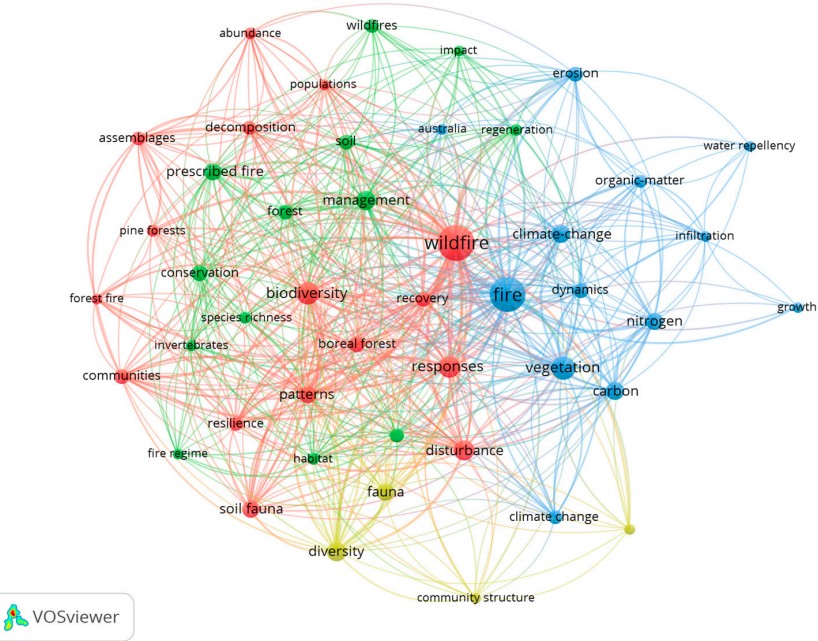

**Figure 12.** High frequency keywords co-occurrence related to the impact of wildland fire on soil fauna.

## 4. Conclusions

In this study we identified and compiled 2388 literature records in the field of wildland fire effects on soil environment, soil microorganisms and soil fauna in the core database of Web of Science, covering the period 1998/1998/1999–2023, and performed bibliometric and visualization analyses using VOSviewer software. The following conclusions can be drawn from the outcomes:

(1)  In terms of the number of publications, research on the effects of wildland fire on the soil environment and soil micro-organisms is at a rapid stage of development, while research on the effects of wildland fire on the soil fauna is still in its infancy.

(2)  The country with the highest number of publications is the United States of America, with the United States Department of Agriculture being the most prolific research institute in this field, and there are many collaborations with other countries and institutions.

(3)  The group of authors in the research field is beginning to take shape, but the group of highly productive and active authors is not sufficiently assembled, and the number of core authors and international teamwork need to be improved.

(4)  The research hotspots focus on the interaction between soil environmental factors carbon, nitrogen, organic matter and soil biodiversity.

We point out that this study is limited to the scope of the Web of Science subject search, and future research can overcome this shortcoming by obtaining more comprehensive data, to provide a more accurate and holistic picture of the current state and trend of research on the effects of wildland fire on the soil environment, soil microorganisms and soil fauna.

## 5. Existing Problems and Perspectives

There are many pressing problems in the research and development of the effects of wildland fire on soils, and the structure of the research system still needs to be improved, which may be reflected as follows:

(1)  The effects of wildland fire on soils may differ, depending on temporal and spatial scales. Since many studies are limited in time and space, long-term and large-scale effects are not comprehensively assessed. The current approach is not sufficient to reveal the dynamic changes over time in different ecosystems after a fire. Future modelling studies addressing ecosystem recovery after a fire may be useful, where fire effects can be modelled using a spatial rather than a temporal approach in order to provide a broader and deeper understanding of the changes and effects of wildland fire on soils.

(2)  The effects of wildland fire on soils may depend not only on the fire itself, but also on a combination of other ecosystem factors, such as vegetation type, soil characteristics and climatic conditions. Comparative studies on a global scale can be conducted in the future, to investigate the differences and similarities of wildland fire on soil environment and soil biota in different regions and ecosystems. This may help to reveal universal patterns as well as specific geographical characteristics.

(3)  At present, the focus is mainly on the determination of hydrothermal factors, total nitrogen and organic carbon. In the future, more soil environmental factors such as ammonium nitrogen, nitrate nitrogen, metal ions, soluble salts, etc., can also be taken into account, to study changes in a wider variety of soil environmental factors and investigate their effects on soil microbial and animal groups after wildland fire, thereby strengthening the in-depth study of soil nutrient cycling.

(4)  Soil biological communities are highly diverse and complex, and different biological groups play different roles in the ecosystem. With the application of molecular biology, gene expression, and the characterization of adaptive mechanisms and functional gene changes of soil biological taxa, biological communities can be further investigated in the future. This will help to understand the molecular response and adaptation strategies of soil microbes and animals after wildland fire.

(5) Evaluating the recovery process of soil biomes after wildland fire requires long-term monitoring, to assess the actual impact of wildland fire on soil biomes. In the future, longer-term and continuously monitored fire trails could be established and controlled experiments could be conducted to isolate and determine the role of specific factors of wildland fire on specific soil microorganisms and soil fauna.

Research into the effects of wildland fire on the soil environment and soil organisms is important to maintain ecosystem stability, combat climate change, protect water resources, manage wildfires and achieve sustainable development. The presented research can provide a scientific basis and guidance for ecosystem management, environmental protection and related policy formulation.

**Author Contributions:** Writing—review and editing, Z.C. and D.W.; visualization, H.P., X.F. and X.L.; supervision, L.Y.; literature search, S.W. All authors have read and agreed to the published version of the manuscript.

**Funding:** Forestry and grassland ecological protection and restoration funds project (GZCG2023-024); Key Research and Development Program of Heilongjiang Academy of Sciences (ZDYF2024ZR02).

**Data Availability Statement:** All data are included in the article.

**Acknowledgments:** Thanks to Wang Pengzhe of Yijia Design Consulting Studio in Baoding, Hebei Province, China, for his help in drawing the graphical abstract.

**Conflicts of Interest:** There are no conflicts of interest to declare.

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
