# Peer review of "Current Status of Research on Wildland Fire Impacts on Soil Environment and Soil Organisms and Hotspots Visualization Analysis"

_fire, doi:10.3390/fire7050163_

Round 1

Reviewer 1 Report

Comments and Suggestions for Authors

This manuscript has been well covered. There is a my comment in this paper. 

Author Response

Thank you for giving us your valuable comments, we have replaced the previous literature as per your request and the replacement literature has been marked in red.

Reviewer 2 Report

Comments and Suggestions for Authors

I read the manuscript with high interest. This work is very actual now, under the global warming and forest fire increasing. This topic is quite relevant in terms of effects of forest fires on the soil environment, on soil microorganisms and soil fauna. In order to provide a baseline for the current research and identify trends on the effects of forest fires, the available literature was identified from the Web of Science database and analyzed. Data were visually displayed for the number of publications, countries, authors, research institutions, and keywords representing research hotspots. The manuscript is well organized and nicely written. The presented results are very important and interesting to readers. The manuscript is well illustrated and very clear.

I would like to thank you for making an important contribution to investigating such a relevant topic.

However, there are some issues to address to, before the manuscript could be accepted.

1.     Abstract, lines 25-26: «The three authors with the largest number of publications are Manuel Esteban Lucas Borja, David A. Wardle, and Konstantin B. Gongalsky». According to text of the article, the three authors with the largest number of publications are David A. Wardle (Swedish Univ. Agr. Sci., Sweden, 19 articles), Pablo Martín-Pinto (Univ. Valladolid Palencia, Spain, 19 articles), Manuel Esteban Lucas Borja (Univ. Castilla-La Mancha, Spain, 16 articles).

2.     Line 80: anIMal – please, correct.

3.     Line 87: Why did you choose this time period – 2007-2023?

4.     Throughout the text: you chose this time period – 2007-2023, but throughout the article You have a lot of references of publications that have been published before 2007.

5.     Line 124: The number of publications describing the effects of fires on the soil environment seemed to be less than on soil microorganisms, is it true? Why do you think this is?

6.     Line 137: I couldn’t find Portugal on the figure 2.

7.     Lines 165-166: there are some extra symbols on the table 1 («С» and «o»)

8.     Lines 202-203: there are some extra symbols on the table 2 («A» and «1»)

9.     Figure 7: What does mean different colours?

10.  Figure 8: this figure is unreadable, please make it larger

11.  Lines 373-385: there are some extra symbols on the table 5

12.  Table 5 and figure 10: the half or more affiliations in table 5 is from Russia, but this country was not marked at figure 10.

13.  Lines 469-470: this statement was not clearly represented in the text. Moreover, according to article the biggest number of publications describes the effects of fires on soil microorganisms.

14.  Lines 527-528: reference No. 8 was not mentioned in the text. Please, check.

15.  Lines 609-610: reference No. 44 was not mentioned in the text. Please, check.

Author Response

Thank you for giving us your valuable comments, we have made changes according to your request, the changes are attached.

Reviewer 3 Report

Comments and Suggestions for Authors

Thank you for this submission to Fire. "Wildfires" greatly impact many ecosystems across the globe, and soils have often been neglected as a topic of interest in wildfire-related research studies. Furthermore, the effects of "prescribed fires" on soil properties and processes have been even more understudied than "wildfires." Therefore, the topic this manuscript attempts to address is appropriate for this journal and the fire community at large. 

However, there are drastically fundamental and fatal flaws in this work, and it must be reframed completely before any consideration of its scientific merit is considered. 

When the term "forest fire" is used broadly to encompass all potential fire effects in natural systems, this is problematic. The term "wildland fire" comprehensively includes "wildfires" and "prescribed fires." Inherently, "wildfires" are the result of ignitions that were not intended as part of a natural resources management plan. Most often, they differ in intensity and severity from prescribed fires that occur in the same landscape. "Prescribed fires" are planned ignitions, and they are often conducted to achieve specific management objectives. Both "wildfires" and "prescribed fires" often occur in non-forested landscapes, such as grasslands, savannas, prairies, meadows, shrublands, etc. Therefore, a paper focused on "forest fires" and their effects on soils must account for: a) landscapes that are only forested and b) both wildfire and prescribed fire effects. Through just a quick search of your paper using the "Find" tool, I see no mention of "prescribed fire," "controlled burning," or "managed fire" in your written text, even though those topics appear in the titles of some of your references. I also see "grassland" and "shrubland" in the titles of some of the papers you reference, but those terms do not appear in your written text. Therefore, your paper is not solely focused on forest fires and it includes information derived from both wildfire- and prescribed fire-related studies, which implies that your title is misleading and scientifically inaccurate. I see that fire intensity is mentioned in the body of your text and you have stated that fire effects often differ based upon fire intensity, frequency, etc. That is accurate, and that should be further explained by differences between "wildfires" and "prescribed fires" that occur in different fuel types because differences in fuel types often alter flammability, rates of spread, heat load, etc. 

I see multiple instances where you have stated "damage" to forests and soil environments, but what does "damage" mean? When used as a verb or noun, "damage" implies an emotion. Scientifically it is a flawed term. Instead, "increase," "decrease," or "alter" should be used in the context of quantifiable differences to the soil environment that wildland fires induce.

Furthermore, the title itself, "Bibliometric analysis of the influence of forest fires on THE soil environment and soil organisms" is not written in proper English. Additional effort must be taken to adhere to proper sentence structure and grammar throughout this paper. Even if the theme of the paper was scientifically sound, which it is not, this paper is not ready for peer-review when evaluated for adherence to proper English. 

If this paper should be rewritten with an emphasis on "wildland fire effects on soil properties and processes," the authors must distinguish and use the proper terminology when referring to fire regimes and fire effects. I would encourage you to read Keeley (2009) to properly frame fire intensity and severity while reading and conducting a new literature review: https://pubs.usgs.gov/publication/70032718.

Comments on the Quality of English Language

Additional effort must be taken to adhere to proper sentence structure and grammar throughout this paper. Even if the theme of the paper was scientifically sound, this paper is not ready for peer-review when evaluated for adherence to proper English. 

Author Response

Thank you for giving us your vital opinion on the article. First, based on your comments, we have rewritten the search terms, redrawn the figures and tables throughout the text, and redescribed them. Second, all references to "damage" to the forest and soil environment in the article have been revised and corrected. Third, we have also reworked the title. Fourth, regarding the classification of fire intensity, I have carefully read the literature you suggested we refer to and determined the writing of light fire, moderate fire, and heavy fire, and corrected the article regarding fire intensity. Fifth, regarding your question about English grammatical structure, we have asked relevant experts to touch up the native language, and the proof of touch-up is attached. The full text of the corrections has been marked in red. Finally, I would like to thank you for pointing out the problems and shortcomings of our article, which made our paper more scientifically meaningful.

Round 2

Reviewer 3 Report

Comments and Suggestions for Authors

Thank you for the attempted revisions that have been included in this version. However, I still feel this version is not ready for publication for 3 primary reasons: 1. Additional attention is needed to improve the written quality of this manuscript. I have included an edited version of the Abstract that should be used as a guide to edit the rest of the text. 2. Your understanding of wildland fire appears flawed based upon the first paragraph of the Introduction. For example, line 58 - what are "light fires?" Are you referring to "low intensity fires?" Intensity, quantified by flame lengths and the generation of heat output, is most often characterized as low, moderate, or high. Severity, most accurately measured as the consumption of soil organic matter as a result of fire, is likewise described as low, moderate/mixed, or high. Therefore, one could assess an individual fire event as low intensity and low severity. Another separate fire could be characterized as high intensity and high severity. As I suggested in the previous comments, you need to read Keeley 2009 to enrich your understanding of fire terminology and fire behavior. After you have a better understanding of how fire behavior is characterized, your descriptions will be more scientifically sound, therefore your results will be more applicable to the complexity and nuances of both wildfires and prescribed fires. As a fire ecologist, I can firmly assert that we do not refer to wildland fires as "light" or "heavy." Furthermore, I can see my advice from my previous critique was not heeded as there is no definitive distinction between "prescribed fire" or "wildfire" in the body of the text. I do not see how these terms can be avoided while writing about this topic. 3. Based upon this work, what would you suggest as future research needs? I see some of this explained in your Conclusions. However, that information is not clearly communicated with a few brief statements in your Abstract. The impact of your work will be stronger if suggestions are made to carry our science forward. 

Comments on the Quality of English Language

This paper continues to struggle with proper sentence structure and verb usage. I have rewritten the Abstract as an example to follow for the rest of the text to improve the written quality of this manuscript.